SES1 is vital for seedling establishment and post-germination growth under high-potassium stress conditions in Arabidopsis thaliana

Guan Peiyan 1
Xie Chen 2
Zhao Dongbo 3
Wang Liyan 1
Zheng Chengchao cczheng@sdau.edu.cn 2
1 Dezhou University , Dezhou , China
2 State Key Laboratory of Crop Biology, Shandong Agricultural University , Tai’an , China
3 Dezhou Academy of Agricultural Sciences , Dezhou , China
Basharat Zarrin
Electronic publication date: 2022 Oct 31
Publication date: 2022
Volume: 10
Electronic Location ID: e14282
Received 2022 May 13; Accepted 2022 Sep 30
Copyright: ©2022 Guan et al.
Copyright year: 2022
Copyright holder: Guan et al.
License: This is an open access article distributed under the terms of the Creative Commons Attribution License, which permits unrestricted use, distribution, reproduction and adaptation in any medium and for any purpose provided that it is properly attributed. For attribution, the original author(s), title, publication source (PeerJ) and either DOI or URL of the article must be cited.
License URL: https://creativecommons.org/licenses/by/4.0/

Keywords: Potassium, SES1, ER stress, Potassium/sodium balance, Arabidopsis

Funding: The Natural Science Foundation of Shandong Province ZR2020QC103 The Doctoral Fund for Scientific Research of Dezhou University 2019XGRC20 The Cultivation Fund for Scientific Research of Dezhou University 2019XJPY02 The Science and Technology Program of Shandong Province for Higher Education J18KA167 This work was supported by the Natural Science Foundation of Shandong Province (No. ZR2020QC103), the Doctoral Fund for Scientific Research of Dezhou University (No. 2019XGRC20), the Cultivation Fund for Scientific Research of Dezhou University (No. 2019XJPY02), and the Science and Technology Program of Shandong Province for Higher Education (No. J18KA167). The funders had no role in study design, data collection and analysis, decision to publish, or preparation of the manuscript.

==============================
Background

The potassium ion (K+) plays an important role in maintaining plant growth and development, while excess potassium in the soil can cause stress to plants. The understanding of the molecular mechanism of plant’s response to high KCl stress is still limited.

Methods

At the seed stage, wild type (WT) and SENSITIVE TO SALT1 (SES1) mutants were exposed to different concentrations of potassium treatments. Tolerance was assayed as we compared their performances under stress using seedling establishment rate and root length. Na+content, K+content, and K+/Na+ ratio were determined using a flame atomic absorption spectrometer. In addition, the expressions of KCl-responding genes and ER stress-related genes were also detected and analyzed using qRT-PCR.

Results

SES1 mutants exhibited seedling establishment defects under high potassium concentration conditions and exogenous calcium partially restored the hypersensitivity phenotype of ses1 mutants. The expression of some K+ transporter/channel genes were higher in ses1-2, and the ratio of potassium to sodium (K+/Na+) in ses1-2 roots decreased after KCl treatment compared with WT. Further analysis showed that the ER stress marker genes were dramatically induced by high K+ treatment and much higher expression levels were detected in ses1-2, indicating ses1-2 suffers a more serious ER stress than WT, and ER stress may influence the seedling establishment of ses1-2 under high KCl conditions.

Conclusion

These results strongly indicate that SES1 is a potassium tolerance relevant molecule that may be related to maintaining the seedling K+/Na+ balance under high potassium conditions during seedling establishment and post-germination growth. Our results will provide a basis for further studies on the biological roles of SES1 in modulating potassium uptake, transport, and adaptation to stress conditions.

Introduction

The potassium ion (K+) is the most abundant inorganic cation in plants and plays important roles in different physiological processes, including turgor regulation, osmotic adjustment, stomatal movements, cell elongation, signal transduction, enzyme activation, and charge carrier (Clarkson & Hanson, 1980; Xu et al., 2006). K+ makes up to 10% of the total plant dry weight (Ashley, Grant & Grabov, 2006; Ragel et al., 2019), and cytosolic K+ concentration is maintained between 80 to 200 mM (Maathuis, 2009). Normal plant growth and development require millimolar K+ in the soil or growth medium (Xu et al., 2006), making it a major plant nutrient. However, K+ concentration at the interface of roots and soils is within micromolar range (Schroeder, Ward & Gassmann, 1994). Additionally, abiotic stress such as salinity or drought has a negative impact on K+ nutrition (Aleman et al., 2014; Nieves-Cordones et al., 2019). K+ deficiency has become a limiting factor for plant growth, leading to decreased crop yield and production. However, its excessive level also leads to distortion of numerous functions in plants; at concentrations above 100 mM, K+ starts to inhibit enzyme activity (Flowers & Dalmond, 1992; Greenway, 1972), destroy the homeostasis of K+ and Na+, and induce salt stress (Li et al., 2021b), so plants have multiple mechanisms for potassium accumulation and release according to its concentration in the environment.

K+ is acquired from the soil through plant roots. In order to receive appropriate K+ nutrients at different concentrations and under different environmental conditions, plant roots have evolved different K+ absorption systems. There are five major families of K+ transporters that have been identified in Arabidopsis according to their structures and functions, including two distinct K+ channel families made up of Shaker-like channels (nine genes), KCO channels (six genes), HKT transporters (one gene), KUP/HAK/KT transporters (thirteen genes), and K+/H+ antiporter homologs (six genes) (Mäser et al., 2001). The Shaker family mediates the major K+ fluxes at the plasma membrane. AKT1, AKT5, AKT6, KAT1, and KAT2 subunits assemble as Kin channels, and the AtKC1 subunit plays as a Ksilent channel, whereas SKOR and GORK subunits form Kout channels. AKT2 is a “weakly rectifying” channel (Very et al., 2014). Kin channels mediate K+ uptake, Kout channels mediate K+ release, and Ksilent channels modify properties of some Kin channels (Dreyer & Uozumi, 2011). High-affinity K+ transporter AtHKT1 does not transport K+ but plays a role in root Na+ transport (Rus et al., 2001; Uozumi et al., 2000). Some KUP/HAK/KT family members that are high-affinity K+ transporters, such as HAK5 and KUP7, uptake in the micromole range of external K+ concentration from the soil, while others may function in both low-affinity and/or high affinity transport (Ragel et al., 2019). The low-affinity K+ channels can uptake K+ at higher external concentrations (>0.1 mM) and the high-affinity K+ transporters can uptake K+ at lower external concentrations (<0.1 mM) (Johnson et al., 2022; Ragel et al., 2019). When the external K+ concentrations are greater than 10 mM, cation/H+ exchangers and cyclic nucleotide gated channels may contribute to K+ absorption (Ragel et al., 2019). However, K+ concentrations higher than 100 mM can destroy the K+ and Na+ homeostasis and induce salt stress (Li et al., 2021b).

Seed germination and seedling establishment are important for the reproductive success of plants, but seeds and seedlings typically encounter constantly changing environmental conditions, such as drought and high salinity (Wang et al., 2020). Before germination, proteins, lipids, carbohydrates and phosphates in the seeds are used as energy sources, and the seeds remain metabolically quiescent (Koornneef, Bentsink & Hilhorst, 2002). When the seed senses a signal conducive to releasing dormancy, the radicle will break through the seed coat and appear, a process called germination (Penfield, 2017). After the post-germination phase lasting 2–3 days, the activation of metabolic signals will lead to the hydrolysis and release of stored substances in seeds. In the process of seedling establishment, if drought and other stress conditions are encountered, seedlings will stop growing and resume growth when environmental conditions are more favorable (Wu et al., 2019). During seed germination, seeds are susceptible to the mother, temperature, nitrate, light, hormones (especially ABA and GA), and different seed tissues (Penfield, 2017). Seed germination and post-germination development are different and interrelated processes with different regulatory mechanisms (Weitbrecht, Muller & Leubner-Metzger, 2011). In the above processes, plants display specific transcriptomes, and these transcriptome changes are usually achieved by activating or repressing a series of different genes. The post transcriptional processes strictly regulate the expression of thousands of genes, as well as the splicing and stability of mRNA. The process of seedling establishment after germination includes cotyledon opening, greening and expansion. This marks the transition of the plant to autotrophic development (Rocío Soledad & Javier Francisco, 2021). ABA prevents seed post-germination growth in order to protect plants from unfavorable conditions (Finkelstein, 2013). ABA-initiated inhibition of seed germination and post-germinative growth is regulated by a core regulatory mechanism that involves the following core components: pyrabactin resistance1, phosphatases type-2C, and sucrose on-fermenting 1-related subfamily 2 kinases (Wang et al., 2020). A complex molecular regulatory network is involved in the process of transition from seed dormancy to germination and from germination to seedling establishment. This network can integrate environmental signals and hormone signals, and achieve this by regulating genes transcription, translation and epigenetic processes. These processes are not fully understood, and more studies are needed to fully elucidate this regulatory mechanisms (Rocío Soledad & Javier Francisco, 2021).

The endoplasmic reticulum (ER) is the site for more than one third of cell protein synthesis and processing (Strasser, 2018), and only properly processed and folded proteins can be exported from the ER and reach their final destinations. As protein folding is an error-prone process, many environmental stimuli, such as salinity, drought, heat, and pathogen infection, often disturb the folding and assembly of newly synthesized proteins, leading to unfolded/misfolded protein overaccumulation and retainment in the ER, resulting in ER stress (Manghwar & Li, 2022). When plants are under ER stress, they turn on ER quality control (ERQC), which includes ER-associated degradation (ERAD), the unfolded protein response (UPR), and autophagy to maintenance of ER protein homeostasis or proteostasis (Chen, Yu & Xie, 2020). Plants can increase the folding capacity of misfolded proteins by up-regulating the expression of molecular chaperone genes, such as BiP, PDI, CNX, CRT, and ERO (Reyes-Impellizzeri & Moreno, 2021).

Previous studies focused on low potassium and NaCl stress, but the molecular mechanism underlying high KCl stress is still limited. Prior studies reported that SENSITIVE TO SALT1 (SES1) encodes an ER-localized molecular chaperone and plays important roles in salt and heat stress (Guan et al., 2018; Guan et al., 2019). The ses1-1 is an EMS-induced mutant with a G-to-A transition at nucleotide 217 within the coding region of AT4G29520 and the ses1-2 is a T-DNA insertion mutant (GABI_944F02). The ses1 mutants exhibited inhibited root growth in plates and exhibited significantly lower survival rate and more severe leaf chlorosis in soil under salt stress. Additionally, ses1 mutants also exhibited sensitivity to high temperatures and had a lower survival rate under heat stress compared with the wild type (WT). In this study, we found ses1 mutants exhibited seedling establishment defects under 125 mM potassium chloride (KCl) conditions. High potassium disturbed the homeostasis of K+/Na+, resulting in ER stress in ses1. This study contributes to our understanding of the SES1 gene related to NaCl and KCl stress and further explores the molecular mechanisms of SES1 in modulating potassium uptake, transport, and adaptation to stress conditions.

Materials & Methods

Plant materials and growth conditions

Arabidopsis plants were from the Columbia-0 background. The two ses1 mutants (ses1-1 and ses1-2) the complementary line (COM2), SES1-overexpressing lines (OE1, OE5), and WT were preserved in our laboratory. COM2 was a complementation line developed by introducing a 3.8-kb WT genomic fragment containing the SES1 promoter, its entire coding region, and 3′ untranslated region into the ses1-2 mutant background. Seeds were surface-sterilized and plated on 1/2 Murashige and Skoog (MS) medium containing 15 g/L sucrose. All media were solidified with 0.8% (w/v) agar. Different concentrations of K+ were supplemented by adding KCl or KNO3. The plates were then kept in the dark at 4 °C for 72 h to synchronize germination, then placed in a growth chamber under long-day conditions with a light/dark cycle of 16/8 h for post-germination growth.

Root growth assay

In order to measure the root length of Arabidopsis thaliana, we plated the WT, ses1-1 and ses1-2 that had been sterilized into a culture plate containing high concentration of K, then placed the plates at 4 °C in the dark for 2 days, after that, the plates were placed in an incubator for two weeks before measuring roots length and taking photographs. The root length is measured using image J 1 software (Wu et al., 2019). Each material in each plate is 10 seedings in total, three biological replicates were set up in the above experiments.

Measurement of K+ and Na+ concentrations in shoot and root

Three-week-old seedlings were treated with 200 mM KCl for 48 h and the shoot and root were harvested, dried for 48 h in an oven at 80 °C until the weight was unchanged, and then ground to powder. Then, 100 mg of shoot and root powder were digested in concentrated nitric acid, hydrochloric acid, and hydrogen peroxide (3:1:1, V:V:V) for 30 min in a microwave 3000 digestion system (Anton Paar, Graz, Austria) for element extraction. Na+ and K+ concentrations were determined using a flame atomic absorption spectrometer (Analytik Jena, Jena, Germany). At the same time, the standard curve was prepared using different concentrations of NaCl and KCl standards, and the absorbance value of the sample was within the range of the standard concentration. The biological duplications of each experiment were repeated four times.

Quantitative real-time PCR (qRT-PCR) analysis

To assay the relative expression levels of related genes, qRT-PCR analysis was performed with the RNA samples isolated from 7-day-old seedlings before and after 200 mM KCl treatment. Total RNA was extracted using TRIzol Reagent (Invitrogen, Waltham, MA, USA) followed by treatment with RNase-free Dnase I (Takara, Kusatsu, Shiga, Japan) at 42 °C for 2 min. The RNA samples (1 µg each) were used as templates for first-strand cDNA synthesis. The total reaction volume for each qRT-PCR was 15 µL, which was comprised of 7.5 µL SYBR Green PCR SuperMix (Takara, Kusatsu, Shiga, Japan), 0.45 µL of each primer, 5 µL of 1:30 diluted cDNA, and 1.6 µL double-distilled water. The PCR reaction conditions were as follows: 95 °C for 30 s and 40 cycles of 95 °C for 5 s followed by 60 °C for 30 s. Reactions were performed using a CFX96 real-time system detector (Bio-Rad). Expression of AKT2 (AT4G22200), KAT1 (AT5G46240), KAT2 (AT4G18290), KC1 (AT4G32650), KUP2 (AT2G40540), KUP3 (AT3G02050), KUP4 (AT4G23640), KCO5 (AT4G01840), SKOR (AT3G02850), GORK (AT5G37500), BiP1 (AT5G28540), BiP3 (AT1G09080), CNX (AT5G61790), DER1 (AT4G29330), ERO1 (AT1G72280), HRD1 (AT1G65040), IRE1A (AT2G17520), and SEL1 (AT1G18260) were identified using the 2−ΔΔCt method. ACTIN7 (AT5G09810) and UBQ10 (AT4G05320) were used as the reference genes. Three biological replicates under similar conditions were performed for each experiment. All primers in this study were synthesized by RuiBiotech and are listed in Table S1.

Results

SES1 positively regulated seedling establishment under high potassium conditions

SES1, an ER-localized chaperone, positively regulated salt stress by alleviating high-salinity induced ER stress, and when the seedlings of ses1 mutants germinated on 12 MS medium were transplanted to 125 mM KCl or KNO3 medium, they showed a short-root phenotype (Guan et al., 2018). However, when seeding WT, ses1 mutants (ses1-1, ses1-2), SES1-overexpressing lines (OE1, OE5), and complementary line (COM2) seeds on 12 MS agar plates with 125 mM KCl or 125 mM KNO3 plates for germination and growth, we found that on 125 mM KCl plates, the ses1-2 mutant barely developed green cotyledons; ses1-1 developed 100% green cotyledons but displayed short-root compared to WT; 100% of OE1, OE5, and COM2 seeds germinated; and seedlings were established as WT. On 125 mM KNO3 plates, ses1-1 and ses1-2 all exhibited severe seedling establishment defects, and most seeds failed to fully emerge from the seed coat. Even ses1-2 did not develop green cotyledons, and ses1-1 developed a green cotyledon ratio of only about 5.5%. Conversely, when grown on medium containing 20 µM K+, ses1-1, ses1-2, and COM2 were indistinguishable from WT plants (Figs. 1A and 1B). In order to exclude the effect of high salt concentration on the osmotic stress of plants, NaCl and NaNO3 were used as a control. The results showed that ses1-2 only displayed short-root compared with the WT on the same concentration of NaCl and NaNO3 medium (Figs. 1A and 1B). High concentrations of sodium did not affect the seedling establishment of the ses1 mutants. These results suggest that SES1 has a significant impact on seedling establishment under high potassium condition.

Figure 1 SES1 positively regulates seedling establishment under high potassium condition.

(A) Image of WT, ses1-1, ses1-2, ses1-2 complementary line (COM2), overexpression lines (OE1 and OE5) seedlings grown on 1/2 MS agar plates with or without corresponding concentration of salt. Photographs were taken after grown horizontally at 22 °C for 14 days. (B) Seedling establishment rate of WT, ses1-1, ses1-2, COM2, OE1, and OE5 in A. Error bars indicate SD (n = 3). ***, P < 0.001 (Student’s t-test).

The ses1 mutants displayed K+ concentration-dependent seedling growth

Because the phenotypes of ses1-1 mutants under 125 mM KCl and 125 mM KNO3 were not similar, we wanted to explore the effect of low concentrations of KCl and KNO3 on these mutants. We performed comparative growth analyses of the WT and ses1 mutant alleles (ses1-1 and ses1-2) subjected to KCl and KNO3 concentrations in the external medium of 40 to 125 mM. After germination and growth on media with different concentrations of KCl and KNO3 for two weeks, their growth status is shown in Figs. 2A and 2B. We found that the ses1 mutants displayed K+ concentration-dependent seedling growth through quantification of their root length. When grown on medium containing 60/80/100 mM KCl or 40/60/80 mM KNO3, ses1 mutants only exhibited short-root length (Figs. 2C–2F). In addition, the root length of ses1-2 was significantly shorter than that of ses1-1 under the same concentration of KCl or KNO3 (Figs. 2C–2F). However, the two independent ses1 mutant alleles displayed the same phenotype, but failed to establish when grown on medium containing concentrations of 125 mM KNO3 (Figs. 2A, 2C and 2E). These results indicate that SES1 plays a vital role in maintaining plant post-germination growth and seedling establishment at a normal level under high potassium concentration conditions, and the mutants displayed K+ concentration-dependent seedling growth.

Figure 2 SES1 displayed K+ concentration-dependent seedling growth.

Growth status of WT, ses1-1, and ses1- 2 at concentrations of 40 mM to 125 mM KNO3 (A) or KCl (B). Photographs were taken after being grown at 22 °C for 14 days. Root length of seedlings in the presence of indicated concentrations of KNO3 (C) or KCl (D) in A and B. Root length of seedlings in C (E) or in D (F) measured by Image J 1. Error bars indicate SD (n = 5).

Exogenous Ca2+ can partially restore the hypersensitive of ses1 mutants under high potassium conditions

Studies have shown that the Ca2+ signaling pathway can regulate the K+ channel and affect potassium uptake and transport in plants (Caballero et al., 2012; Li et al., 2006), so we examined the effect of exogenous calcium on the hypersensitivity of ses1 mutants under high potassium conditions. We found when adding 5 mM CaCl2 to 125 mM KNO3 medium, the seedling establishment rate of ses1-1 and ses1-2 and their hybrid lines F1 (1- 1 ×1-2) increased, but CaCl2 had no effect on the seedling establishment rate and sodium-sensitive phenotype of ses1 mutants (Figs. 3A and 3B). These results indicate that exogenous Ca2+ can partially restore the seedings establishment deficiency caused by high K+.

Figure 3 Ca2+ can partially restore the hypersensitivity of ses1 mutants under high potassium conditions.

(A) Image of WT, ses1-1, ses1-2, and se1-1 ×1-2 (F1) seedlings grown on 125 mM KNO3 or NaNO3 plates with or without 5 mM or 8 mM CaCl2. Photographs were taken after being grown horizontally at 22 °C for 14 days. (B) Seedling establishment rate of WT, ses1-1, ses1-2, and F1 in A. Error bars indicate SD (n = 3).

Different transcription levels of KCl-related genes in ses1-2 and WT

K+ is the most abundant cation in plants and is necessary for cell growth, and cytosolic K+ concentration is maintained between 80 to 200 mM (Maathuis, 2009). The ses1 mutants displayed K+ concentration-dependent seedling growth and establishment, and we theorized that the transcription levels of K+ transporter genes may be different in ses1 and WT. To test our hypothesis, we detected some K+ transporters such as the Shaker superfamily of potassium channels, which included inward-rectifying channels (KAT1 and KAT2), a weakly rectifying channel (AKT2), Shaker-like protein (KC1), outward-rectifying channels (SKOR and GORK), and “two-pore” K+ channel KCO5, KUP/HAK/KT family transporters (KUP2, KUP3, and KUP4). As shown in Fig. 4, we detected much higher transcription levels of these genes (except for KUP3 and KC1) in ses1-2 mutants, especially under KCl treatment. The different expression levels of K+ transporters genes in ses1 mutants and WT before and after 200 mM KCl treatment, indicate their potential important functions and signaling transduction pathways.

Figure 4 Altered expression of K+ transporter genes in ses1-2 after 200 mM KCl treatment for 6 h.

One-week-old seedlings before and after 200 mM KCl treatment were harvested for RNA extraction and qRT-PCR analysis of the relative expression of K+ transporters. WT plants were used as a control and the relative expression of UBQ10 used ACTIN7 as an internal standard. ACTIN7 and UBQ10 were used as an internal standard for the relative expression of K+ transporter genes.

Potassium sensitivity of ses1 may be associated with the disturbance of an optimal sodium/potassium ratio in root

In order to characterize the underlying potassium-sensitive mechanisms of ses1 mutants, the K+ and Na+ contents in WT and ses1-2 plants before and after 200 mM KCl treatment were measured. In the root, the K+ content in WT increased significantly after high potassium treatment, but in the ses1-2 mutant, K+ content in the root had no significant difference before and after high potassium treatment. In addition, under normal conditions, there was no significant difference in K+ content between the ses1-2 mutant and the WT roots. In the shoot, under normal conditions, the K+ content of the ses1-2 mutant was significantly higher than that of the WT. After high potassium treatment, the K+ content of both significantly increased, but there was no significant difference between the two (Figs. 5A and 5B). However, the Na+ content of root was significantly higher in ses1-2 than WT before and after treatment with 200 mM KCl for 48 h, resulting in a significantly decreased K+/Na+ ratio of root in ses1-2 compared with WT (Fig. 5A). Meanwhile, the Na+ content and K+/Na+ ratio of shoot showed no differences between ses1-2 and WT plants (Fig. 5B). These results indicate that the ses1-2 mutant can maintain stable K+ levels in root better than WT, which leads to the K+/Na+ ratio in the root of ses1-2 mutant being significantly decreased compared to WT, and this might be the reason why ses1-2 mutants are potassium sensitive. SES1 may be related to maintaining a stable seedling sodium/potassium balance under high potassium conditions.

Figure 5 Changes in K+, Na+ content, and K+/Na+ ratio in WT and ses1-2.

(A) Total K+, total Na+ content, and K+/Na+ ratio in WT root and ses1-2 before and after KCl treatment. DW indicates dry weight. The bars indicate the mean ± SD of three independent measurements. ns indicates no significance. **, P < 0.01 (Student’s t-test). (B) Total K+, total Na+ content and K+/Na+ ratio in WT shoot and ses1-2 before and after KCl treatment. Error bars indicate SD (n = 3). ns indicates no significant differences (Student’s t-test).

ER stress-related genes’ transcription level was higher in ses1-2 mutant

High concentrations of salt can induce ER stress (He et al., 2021). Therefore, we speculated that the hypersensitivity of ses1 mutants to potassium probably resulted from severe ER stress. To test our hypothesis, we determined the transcription levels of ER stress-responsive marker genes, including BiP1, BiP3, CNX, DER1, ERO1, HRD1, IRE1A, and SEL1, in WT and ses1-2 mutants. We found that ER stress-related genes had much higher transcription levels in ses1-2 compared with WT after KCl treatment (Fig. 6). These results revealed that ses1-2 suffers more serious ER stress than WT, and ER stress may influence the resistance of ses1-2 under high KCl conditions.

Figure 6 ER stress-related genes showed higher expression in ses1-2.

One-week-old seedlings before and after 200 mM KCl treatment were harvested for RNA extraction and qRT-PCR analysis the relative expression of ER stress related genes. WT plants were used as a control. ACTIN7 and UBQ10 were used as an internal standard for the relative expression of ER stress-related genes.

Discussion

Although potassium is necessary for plant growth, K+ at concentrations higher than 50 mM can disturb Na+/K+ homeostasis and cause the ion toxicity (Aghajanzadeh et al., 2018). When the stress of NaCl and KCl were at the same concentrations, KCl stress caused more serious damage to plants than NaCl stress (Yao et al., 2009). In our study, we found that when under 125 mM NaCl, the phenotype of ses1 only showed short-root length, while when under 125 mM KCl, ses1 did not develop green cotyledons like WT, and ses1-2 displayed K+ concentration-dependent seedling growth (Figs. 1 and 2). After 200 mM KCl treatment, large amounts of K+ flowed into the cytoplasm and caused an imbalance between K+ and Na+. The K+ contents in root and shoot were sharply induced in WT, while in the root of ses1-2, K+ did not change sharply. The ses1-2 mutant could maintain a stable K+ level in root better than WT, which led to the K+/Na+ ratio in the root of ses1-2 mutant significantly decreasing compared to WT (Fig. 5). These results indicate that SES1 could affect ion transport under KCl stress. These results showed that the expression levels of genes related to potassium transfer also changed before and after KCl treatment in WT and ses1-2 (Fig. 4). Ca2+ is a very important secondary messenger. Under abiotic stresses, especially salt stress and heat stress, Ca2+ bursts in the cytoplasm, thus exerting its messenger function to regulate plant signal transduction pathways (Ding et al., 2010). External calcium in plants facing salinity may be associated with the maintenance of an optimal sodium/potassium ratio in the cytosol (Shabala et al., 2006), which is consistent with our results that showed that Ca2+ can partially restore the hypersensitivity of ses1 mutants under high potassium conditions (Fig. 3). Although high concentrations of salt, such as NaCl and KCl, can induce ER stress, these results indicate that the molecular mechanisms of SES1 in modulating plant response to these two stresses may be different. The current studies focus on NaCl stress, and the molecular mechanism underlying KCl stress is still unclear. The sensitive phenotype of ses1-2 on KCl medium provides good material for studying the molecular mechanism under high potassium stress and ER stress (Fig. 6). In the phenotype experiment, the potassium ion concentration varies from 40–125 mM, while in the gene expression and ion determination experiments, we used 200 mM potassium ion treatment. What is the difference between the two? We think that in phenotype experiment, the phenotypic changes of the mutant and the wild-type were obvious after two weeks of growth at the concentration of 125 mM K+, but in the gene expression and ion determination experiments, because the treatment time is only a few hours, so we used a higher concentration of potassium ion treatment. We think that the effects of “low concentration + long time” and “high concentration + short time” on seedlings are similar, and we think that this is more reasonable.

Under KCl stress conditions, plants are typically stressed in three ways: ionic stress, osmotic stress, and oxidative damage (Li et al., 2021a; Li et al., 2021b; Zheng et al., 2021). We found that SES1 can maintain the seedling K+/Na+ balance, and the expression of K+ transporter genes and ER stress-related genes were altered between WT and ses1 mutants under high potassium conditions during seedling establishment and post-germination growth. Seed priming with KCl was an effective method for alleviating seed germination caused by salt stress (Ben Youssef et al., 2021) and improve emergence and seedling establishment under stress (Ella, Dionisio-Sese & Ismail, 2011; Haider & Rehman, 2022) through a changed K+/Na+ratio. While high K+ concentrations could also induce salt stress and inhibit seedling establishment and post-germination growth of plants (Bakshi et al., 2018; Hassini et al., 2017), perhaps through plant hormone signal transduction and the mitogen-activated protein kinase signaling pathway (Bakshi et al., 2018; Li et al., 2021b). We could detect the ABA and GA contents of WT and ses1 mutants before and after KCl treatment during seedling establishment, and differentially-expressed genes using transcriptome to reveal SES1’s mechanism of regulating establishment and post-germination growth in the future. We tried to screen the reciprocal protein of SES1 using the yeast two-hybrid and protein profiling method, but since it is a protein localized on the ER, neither of these two methods screened the reciprocal protein. In the future, we may try other methods to search for the mechanism of SES1 regulation of potassium stress in plants. We noted that potassium-sensitized yeast is a powerful genetic tool that has been leveraged in several different studies (Mackie & Brodsky, 2018). Both Nha1 and Ena1 promoted strain growth in high-K+ medium, while a strain lacking both the ENA cluster (ena1-4 Δ) and NHA1 failed to propagate on high concentrations of potassium (BaAueIos et al., 1998; Kinclova et al., 2001). In future studies, we can introduce SES1 into the ena1-4 Δ nha1 Δ yeast strain and observe whether the strain can recover phenotypically on high potassium media and clarify whether SES1 is associated with potassium export.

Conclusion

Based on our findings, we established a simplified working model for SES1 under KCl stress (Fig. 7). When K+ stress occurred, Na+/K+ homeostasis was disturbed, and ses1-2 suffered more severe ER stress. These two aspects may have contributed to the fact that ses1-2 exhibited a sensitive phenotype under high potassium conditions.

Figure 7 A simplified working model for SES1 under KCl stress.

When K+ stress occurred, Na+/K+ homeostasis was disturbed, and ses1-2 suffered more severe ER stress. These two aspects may have contributed to the fact that ses1-2 exhibited a sensitive phenotype under high potassium conditions.

Supplemental Information

Supplemental Information 1 The original images and data for Fig. 1 and 2

Click here for additional data file.

Supplemental Information 2 The original images and data for Fig. 3 to 7

Click here for additional data file.

Supplemental Information 3 Primers used in this study

Click here for additional data file.

Additional Information and Declarations

Competing Interests

Author Contributions

Data Availability

The authors declare there are no competing interests.

Peiyan Guan performed the experiments, analyzed the data, prepared figures and/or tables, authored or reviewed drafts of the article, and approved the final draft.

Chen Xie performed the experiments, analyzed the data, prepared figures and/or tables, authored or reviewed drafts of the article, and approved the final draft.

Dongbo Zhao analyzed the data, prepared figures and/or tables, and approved the final draft.

Liyan Wang analyzed the data, prepared figures and/or tables, and approved the final draft.

Chengchao Zheng conceived and designed the experiments, authored or reviewed drafts of the article, and approved the final draft.

The following information was supplied regarding data availability:

The raw data are available in the Supplemental Files.

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
