# Peer review of "SES1 is vital for seedling establishment and post-germination growth under high-potassium stress conditions in Arabidopsis thaliana"

_PeerJ, doi:10.7717/peerj.14282_

## Round 0.1 · original submission · Major Revisions

Kindly address the points raised by the reviewers.

Reviewer 1 ·

Basic reporting

no comment

Experimental design

no comment

Validity of the findings

no comment

Additional comments

In this manuscript, the authors found ses1 mutants exhibited seedling establishment defects under 125 mM potassium chloride (KCl) conditions. High potassium disturbed the homeostasis of K+/Na+ then results in ER stress in ses1. They further explored the molecular mechanisms of SES1 in modulating potassium uptake, transport and adaptation to stress conditions. This manuscript may attract some readers who have interests in K+ and salt stress in plants. However, there are some essential issues should be solved:

1. The ses1 muants sources should be stated, as this is an independent publication, the readers won’t try to trace what’ it is.
2. What are the criteria for a seedling’s “establishment” and “photomorphogenesis” in this study?
3. Since the manuscript describe the “establishment and postgermination growth”, the correlated background knowledge i.e. the published regulatory mechanisms should be briefly reviewed in the introduction part. Correspondently, how the new findings are related to the known ones should be discussed.
4. Why the phenotyping works are conducted at maximum 125 mM K+, while the gene expression and ion determination are at 200? What’s the difference?
5. Line 157, “same phenotype”? do you mean similar or comparable? The English language should be improved to ensure that an international audience can clearly understand your text.
6. About 3 mM Ca2+ are included in MS medium, but no obvious effect in mutants under 125 K+ stress. This at least demonstrate a dose-effect for Calcium on restoring toxicity. Why the author did not try higher concentrations?
7. In the conclusion part, “… SES1 is a major physiologically relevant molecular may be related to maintaining the seedling K+/Na+ balance under high potassium conditions …”. I don’t think so, your manuscript only presents some proof that ses1 is involved in K+ toxics, but far from “major physiologically relevant molecular”. I suggest you tone down a little bit for your findings.

Minor part:
8. “Image J software”, no citation, no version number.
9. Cite Table S1 in appropriate place in M&M
10. How statistics performed are not mentioned in M&M.

Reviewer 2 ·

Basic reporting

The authors have done a great job. The English writing is clear and professional. the manuscript is well organized and experiments well described.
while most raw data is supplemented, just one minor edit,
1. I suggest that the authors should also provide the genotype of the mutants and stains being used in the method section.

Experimental design

the authors did this part very well.

Validity of the findings

the authors did this part very well.

Reviewer 3 ·

Basic reporting

· The manuscript contains a lot of grammatical errors, and the language is not scientific enough in general. I pointed some of them out here, but I suggest the manuscript should be carefully edited throughout.

1. Line 15; Change “The molecular mechanism of plants response to high KCl stress is still limited.” to “The understanding of the molecular mechanism of plants response to high KCl stress is still limited.”
2. Line 18; Change “Tolerance was assayed as the performance of stressed...” to “Tolerance was assayed as their performance under stress…”
3. Line 23; Change “…under high potassium conditions” to “under high potassium concentration conditions”
4. Line 24; Delete “and these defects were K+ concentration-dependent” because the previous sentence already covers it.
5. Line 32; Change “molecular” to “molecule”.
6. Line 42; Change “and” to “to”.
7. Line 51; Change “it” to “its”.
8. Line 65; Change “numbers” to “members”
9. Line 179; Change “toxicity” to “deficiency”.
10. Line 201; Delete “extremely”.
11. Section 3.6; Change all the “expression” to “transcription”.

· I suggest that the introduction should be updated to include some extra background details. My suggestions are as follows:

1. Line 86-93; I suggest adding a brief introduction about ses1-1 and ses1-2 mutants here. I am aware that both mutants have been described in the author’s previously published paper (Guan et al. 2018), but because this study is mainly focused on evaluating the mutants, it will be helpful for readers to have a background understanding of them in order to follow the discussion.

Experimental design

No comment here, experimental design in this manuscript is good.

Validity of the findings

Please consider the revising suggestions I listed below.


1. Line 141; I didn’t find a clue for what is the “ses1-2 complementary line (COM2)”. Better to clarify the difference between it and the ses1-2 mutant.
2. Figure 1; In figure 1A, for the “125mM KCl” condition, the seedling establishment of OE1, OE5 and COM2 are apparently better than WT. For the “125mM KNO3” condition, OE1 and OE5 are better than the WT. Based on that observation, I suggest reassessing the percentage of seedling establishment for the aforementioned lines in figure 1B accordingly.
3. Line 144; Suggest changing “100%” to “150%”, the reason is stated in #13.
4. Line 151; I didn’t see the difference in root length between ses1-1 and WT under NaCl and NaNO3 treatment, suggest deleting the conclusion for ses1-1 here.
5. Line 155; The conclusion is not correct. I suggest changing to “ses1 mutants displayed…”
6. Line 162-164; The figure 2D didn’t convince me that “ses1 mutants” exhibit a short-root length under <100mM KCl condition. This conclusion is only applicable to “60mM KCl”. Authors should qualify their statements appropriately.
7. Line 168-169; I don’t think this conclusion properly summarizes the findings in section 3.2. Suggest changing to “These results indicate that SES1 plays a vital role in maintaining plant postgermination growth and seedling establishment at a normal level under high potassium concentration condition, and …”
8. Line 176; Based on the observation of figure 3A, I don’t think either the seedling establishment rate of ses1-2 or their hybrid lines F1 (1-1×1-2) are “significantly” increased. Authors should qualify their statements appropriately.
9. Line 180; Change “3.4 Expression … ses1 and WT” to “3.4 Transcription … ses1-2 and WT”. Since the study in section 3.4 only covers the study of the transcription levels of KCl-related genes in mutant ses1-2, only saying ses1 here is misleading.
10. Line 184; Suggest change “…expression levels of K+ transporters maybe…” to “transcription level of K+ transporter genes maybe…”. Because qRT-PCR can only analyze the gene transcription at RNA level. “Expression” should more likely be used in “protein level”.
11. Figure 4; Should include the ACTIN7 and UBQ10 control data here.
12. Line 189; In figure 4, not only is the KUP3 an exception, but also the KC1 transcription level getting decreased in ses1-2 mutant before the KCl treatment. Should make a more rigorous statement in the main text.
13. Section 3.4; The data in figure 4 are showing that, even under the low K+ condition, the transcription level of most of the KCl-related genes got significantly increased in ses1-2 mutant. That’s suggesting the elevation of the transcription level of those genes is more like an intrinsic property of ses1-2 mutant, not a “response to KCl stress”. The author should reassess the conclusion here.
14. Line 199; The observation here is wrong. The K+ content under high-KCl treatment didn’t decrease in ses1-2 in roots, on the contrary, it’s more like the WT K+ content significantly increased compared with CK.
15. Line 204-207; I don’t think the data in figure 5 is supporting the conclusion here and I’m actually got a bit confused by those data. To me, it’s more likely that the ses1-2 mutant can maintain the K+ level in root stable better than WT, which leads to the K+/Na+ ratio in the root of ses1-2 mutant significantly decreased than WT, and that might be the reason why ses1-2 mutant is potassium sensitive. I suggest the author to revisit the data in Figure 5 and to make a more solid conclusion here.
16. Line 208; Change “…related genes were higher in…” to “…related genes transcription level was higher in…”

---

## Round 0.2 · Minor Revisions

Criteria for a seedling’s “establishment” and “photomorphogenesis” in this study is still not appropriate. Please add clear information in the manuscript and reproduce text in rebuttal letter.

Authors say that they have revised the introduction sections to include contents related to establishment and post-germination growth of seedings and its regulation mechanism in line 75-103. Correspondently, new findings related to the known ones in line 290-303. However, I am unable to see new text related to these points at mentioned line numbers. Please reproduce text with rebuttal for clarity.

Why the phenotyping works are conducted at maximum 125 mM K+, while the gene expression and ion determination are at 200? What’s the difference?
Author answer: Because under the concentration of 125 mM K+, some wild-type seedlings can still grow normally. If the concentration of K+ is increased to 200 mM, almost all seedlings cannot grow normally and die completely. The phenotypic changes of the mutant and the wild-type were obvious after 2 weeks of growth at the concentration of 125 mM K+. In addition, in the determination of gene expression and ion concentration, seedings were only treated for 6 hours. Because seedings were treated for a short time, a high concentration, 200 mM potassium ion, was used.

This should be added in manuscript as well.

---

## Round 0.3 · accepted · Accept

Authors have now satisfactorily replied to all the points raised by the editor and reviewers. The criteria for seedling establishment and photomorphogenesis is now clear and the rebuttal includes addressing of key points, the addition of new literature, etc. The manuscript can now be accepted.